# OCTOPUS: AN AUTO-GENERATED MULTIDIMENSIONAL FINE-GRAINED BENCHMARK FOR EVALUATING TEXT-TO-SQL SYSTEMS

## ABSTRACT

Text-to-SQL is to convert natural language queries into structured SQLs, facilitating user interaction with databases without any SQL knowledge. The advent of LLM technologies significantly accelerates the text-to-SQL development. It is important to construct an appropriate benchmark to evaluate the performance of text-to-SQL models. However, existing text-to-SQL benchmarks are mainly produced by human annotations and suffer from limitations of low SQL complexity, single questioning mode, and low scalability. To address these limitations, we present a new multidimensional text-to-SQL benchmark, called OCTOPUS, which contains comprehensive evaluation metrics and fully auto-generated datasets. OCTOPUS has 9 first-level metrics and 18 second-level metrics from four dimensions to evaluate the performance of text-to-SQL systems, including accuracy, robustness, interactivity, and generalization. To help the benchmark construction, we also propose a series of fully automatic text-to-SQL data generation methods, which reduce human involvement, improve efficiency, and support higher scalability. OCTOPUS consists of 10,885 complex question-SQL pairs and 10,874 multi-turn dialogues over 74 public databases. We evaluate state-of-art text-to-SQL models on OCTOPUS and find they have unsatisfactory performance in all testing metrics and are still far from practical applications. OCTOPUS can be used to enhance the accuracy and utility of text-to-SQL models.

## 1 INTRODUCTION

Text-to-SQL is an active research area at the intersection of natural language processing (NLP) and database management, aiming at bridging the gap between human language and database queries (Gkini et al., 2021; Kim et al., 2020; Katsogiannis-Meimarakis & Koutrika, 2023; Li et al., 2024a). Text-to-SQL enables users to retrieve or manipulate database through natural language queries even without any SQL knowledge. Recently, it has been witnessed a significant advancement of text-to-SQL fueled by the breakthroughs in large language models (LLMs) (Fu et al., 2023; Fan et al., 2024; Gu et al., 2023; Gao et al., 2024). According to the SPIDER leaderboard[1], a well-known benchmark for text-to-SQL, the top-performing model has achieved an execution accuracy of 91.2%, and many other solutions integrating LLMs have generally reached accuracies above 80%. Despite of the considerable advancements, there are still many challenges, including 1) the lack of comprehensive and fine-grained benchmarks that accurately reflect the complexity of real-world queries, 2) poor performance of text-to-SQL systems on handling ambiguous, complex and domain-specific queries, and 3) the difficulties of integrating of text-to-SQL models into user-friendly applications. Nowadays, the research on text-to-SQL has shifted towards addressing limitations of existing text-to-SQL models in real-world applications where models are required to generate more complex SQL statements and handle ambiguous or incomplete user queries (Sen et al., 2020; Guo et al., 2019; Deng et al., 2021; Wang et al., 2023).

However, existing benchmarks are not sufficient to satisfy the above evaluation requirements due to the discrepancies of the distribution between test data and real-world data. They fail to offer fine-grained targeted evaluation or provide comprehensive and multi-dimensional evaluation

---

[1]https://yale-lily.github.io/spider

Table 1: Overview of our benchmark metrics

| Test Dimension | First-level Metrics | Second-level Metrics |
|---|---|---|
| Accuracy | Database Complexity | Field Naming Complexity |
| | | Table Similarity |
| | | Table Coupling Degree (Minimum number of tables that need to be involved in a query) |
| | Gold SQL Complexity | SQL Structural Complexity |
| | | SQL Operation Diversity |
| | NL (Natural Language) Question Diversity | Diversity in NL Questioning Ways |
| | | Ambiguity of NL Questions |
| | Logical Reasoning Complexity | Number of Reasoning Steps Required to Obtain the Query Result from the Question |
| | External Knowledge Complexity | Variety of External Knowledge Required for Generating SQL from Questions |
| | | Number of Items of External Knowledge Required for Generating SQL from Questions |
| Robustness | Confusion Question Testing | Questions in the Dataset Including Everyday Conversations (Non-SQL Q&A) |
| | | Questions in the Dataset Including Ambiguous Questions |
| | | Questions in the Dataset Requiring Querying Information Outside the Database |
| | | Questions in the Dataset Including Queries Not Supported by SQL Statements |
| | Perturbation Testing | Including Perturbations to the Database |
| | | Including Perturbations to Natural Language Questions |
| Interactivity | Multi-turn Q&A | Dataset Containing Multi-turn Interactive Q&A |
| Generalization | SQL dialect diversity | Dataset Containing SQL Statements in Multiple Database Languages |

metrics. For instance, the targeted SQLs of SPIDER are relative simple and based on small-scale databases, whereas real-world SQL queries are more complex and often based on large-scale databases. BIRD (Li et al., 2024b) enhances database complexity in SPIDER by incorporating large databases, domain knowledge, and a new evaluation metric about the execution speed. However, it still lacks consideration for the diversity and ambiguity of user queries, lacks detailed categorization and definition of domain knowledge types, and suffers from insufficient diversity in SQL operations and high generation cost based on manual generation. SCIENCEBENCHMARK (Zhang et al., 2024b) proposed a semi-automatic text-to-sql dataset generation method based on manual generated seed dataset and generated a challenging benchmark over three domain-specific databases. Although SCIENCEBENCHMARK is more closer to the real application scenario for the professional domain and improve scalability of the benchmark through semi-automatic generation method, it still needs SQL experts' effort to generate seed data and the diversity of generated samples is limited by the seed data.

To address the above limitations of the existing benchmarks, we propose a novel benchmark designed to evaluate the multi-dimensional capabilities of text-to-SQL systems. Based on four performance dimensions of text-to-SQL systems—accuracy, robustness, generalization, and interactivity, we design 9 first-level metrics and 18 second-level metrics to formulate our benchmark. To facilitate our benchmark construction and improve scalability, we also design and implement a series of question-SQL pair generation algorithms and pipelines for the automatic construction of our benchmark. Our benchmark consists of 10,885 complex question-SQL pairs and 10,874 multi-turn dialogues over 74 public databases, covering 9 first-level metrics and 18 second-level metrics. We evaluate state-of-art text-to-SQL models on OCTOPUS and find they have unsatisfactory performance in all testing metrics and are still far from practical applications. The evaluation results show the importance of OCTOPUS to be proposed. OCTOPUS can be used to enhance the accuracy and utility of text-to-SQL models. The main contributions of our paper are summarized as follows:

## 2 RELATED WORK

To improve the accuracy of text-to-SQL models and promote the practical application of text-to-SQL systems, many famous benchmarks have been proposed in recent years. SPIDER (Yu et al., 2018) is the first well-known benchmark to introduce a cross-domain dataset containing SQL queries of varying difficulty levels and to propose two evaluation metrics for measuring the accuracy of text-to-SQL models. Based on SPIDER, the DR.SPIDER benchmark (Chang et al., 2023) provides a comprehen-

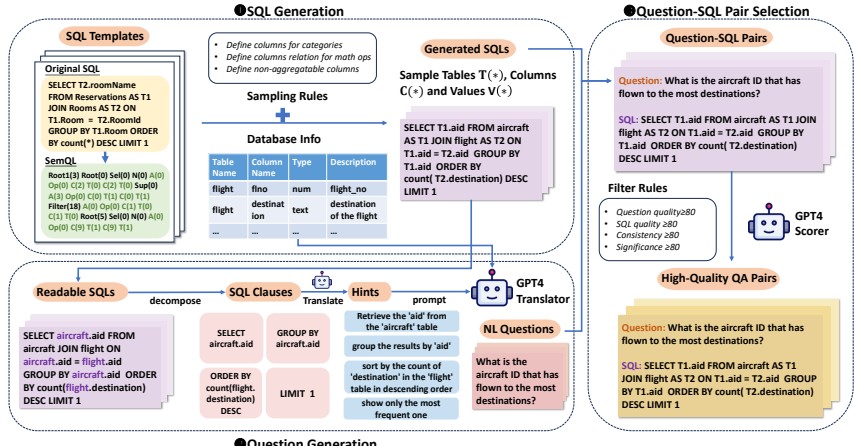

Figure 1: Overview of the pipeline for question-SQL pairs automated generation

sive evaluation framework for assessing the robustness and generalization across diverse domains and complex SQL queries. BIRD (Li et al., 2024b) incorporates real-world large-scale databases and more complex SQL queries, and further emphasizes the impact of noisy database values and external knowledge. In addition to the above benchmarks that focus on single-turn question-SQL pair, some works focus on evaluating the ability of text-to-SQL systems to engage in multi-turn conversations. SPARC (Yu et al., 2019b), CHASE (Guo et al., 2021) and COSQL (Yu et al., 2019a) extend question-SQL pairs from SPIDER to multi-turn dialogues, thereby building cross-domain corpus with different conversation types and languages. Our benchmark is the first automatically generated multidimensional benchmark covering fine-grained metrics to evaluate the comprehensive ability of text-to-SQL models.

## 3 BENCHMARK DESIGN

From the perspective of practical applications, we summarize and formulate four dimensions of metrics to evaluate text-to-SQL systems: *accuracy, robustness, interactivity* and *generalization*. We further divide each dimension into more fine-grained metrics with two levels. We design and develop our benchmark following these metrics to ensure a comprehensive evaluation of the text-to-SQL system's capabilities across different facets. The overview of our benchmark metrics is listed in Table 1. We will briefly introduce the benchmark metrics in the following due to the page limit. The detailed definitions of our benchmark metrics is described in Appendix A.

**Accuracy**   Ensuring the correctness of generated SQLs is the primary requirement of text-to-SQL system. The difficulty of generating SQL is affected by many factors in real applications. For testing the accuracy of text-to-SQL systems, we formulate five first-level evaluation metrics, including *Database Complexity*, *Gold SQL Complexity*, *Natural Language Question Diversity*, *Logical Reasoning Complexity* and *Domain Knowledge Complexity*. These metrics involve the major challenges faced by text-to-SQL systems when generating correct SQLs and require the benchmark to contain targeted test samples that meet the above metrics.

**Robustness**   A stable text-to-SQL system needs to be robust to disturbances and changes from internal and external sources. We aim to test the robustness of the text-to-SQL system from two following aspects: *Confusion Question Testing* and *Perturbation Testing*. These two metrics require the benchmark samples simulate perturbations that occur in real applications to evaluate the robustness of text-to-SQL systems.

**Interactivity**   A user-friendly text-to-SQL system needs to handle variable user queries. The metric in this dimension requires the benchmark dataset to contain multi-turn question-SQL pairs in different conversations to test the text-to-SQL systems' ability to handle multiple rounds of dialogue. Multi-turn Q&A samples in the text-to-SQL domain should closely mimic the user's question pat-

terns observed in real applications. Additionally, they must precisely present the correct response behavior of text-to-SQL systems.

**Generalization** A general text-to-SQL system needs to have the ability to support multiple types of database systems. This metric requires the benchmark to contain gold SQLs in multiple SQL dialect formats to test the generalization ability for different database systems. *SQL dialect diversity* can be measured by the number of SQL dialects. To the best of our knowledge, this metric has not been mentioned in previous benchmarks, and our paper is the first to propose it.

## 4 BENCHMARK CONSTRUCTION

### 4.1 DATABASE COLLECTION

To simulate real-world scenarios as much as possible for testing a text-to-SQL system, we manually collected 74 real-world complex databases from the internet. The source of databases includes MySQL example databases[2] (7%), WikiDBs[3] (7%), CTU Prague Relational Learning Repository[4] (27%), Kaggle[5] (32%), and SPIDER benchmark (Yu et al., 2018) (27%). MySQL official website provides 5 high-quality and large-scale example databases to test the functionality of MySQL database system, which is collected to construct our benchmark. WikiDBs (Vogel & Binnig, 2023) is a novel open-source corpus of 10,000 relational databases collected from Wikidata, each of which consists of multiple tables connected by foreign keys. We manually select complex databases with at least 8 tables from WikiDBs. The CTU Prague Relational Learning Repository is an open platform for machine learning with multi-relational data which hosts 50 databases. We select top 20 complex databases from CTU. Also, we sampled 20 databases from SPIDER with consideration of intersectionality. The

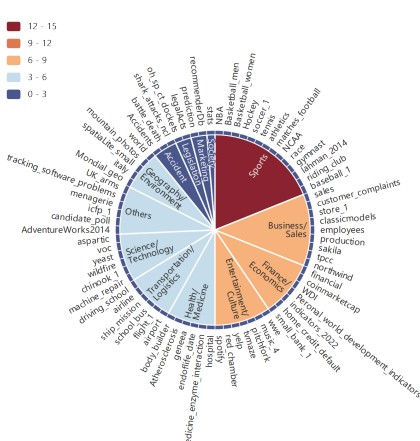

Figure 2: The domain distribution of benchmark databases

rest of databases are selected from Kaggle, which contains high-quality dataset for data science competitions and collaboration. In our database collection, we specifically select large and complex databases, characterized by numerous tables, extensive fields, and intricate foreign key constraints, to closely simulate real-world enterprise databases. Our 74 public databases reaches 23 GB, and covers a wide range of specific domains, such as sports, finance, medicine, retail, etc.

### 4.2 DATASET CONSTRUCTION

In previous famous benchmarks (Li et al., 2024b; Yu et al., 2018), question and SQL generation is completed by human writers and annotators expert in text-to-SQL. Obviously, it is often time-consuming and labor-intensive if relying only on manual generation. To effectively generate question-SQL pairs and improve scalability, we propose a new question-SQL pair generation pipeline which can fully automate the generation of SQLs along with corresponding questions over any database. As shown in Figure 1, our fully automated generation pipeline is composed of three main parts: SQL generation, Question generation, and Question-SQL pair selection. The following will describe the details of our generation algorithm.

**SQL Generation:** We integrate the SQL generation technique from SCIENCEBENCHMARK (Zhang et al., 2024b) with SQL templates derived from real-world applications to automate SQL

---

[2]https://dev.mysql.com/doc/index-other.html

[3]https://wikidbs.github.io/

[4]https://relational-data.org/

[5]https://www.kaggle.com/

generation. We first construct a SQL template library comprising SQL templates with various structures. To obtain SQL templates, we collect SQLs from the internet, previous benchmarks, and enterprise applications, then convert them into an Abstract Syntax Tree (AST) representation called SemQL (Guo et al., 2019). In these SQL templates, let the three placeholders $\mathbf{T}(*)$, $\mathbf{C}(*)$, and $\mathbf{V}(*)$ replace tables, columns, and values related to the database respectively. To generate SQLs on a specified database, the sampling algorithm takes configured sampling rules and the database information as input and then fills the placeholders with concrete values according to the specified database. The sampling rules specify the scope and priority of tables and columns to ensure semantic correctness. Intuitively, the closest table measured by foreign key connections will be sampled at a high priority. Columns are labeled with three types: categorical columns, computable columns, and non-aggregatable columns. The column types are automatically annotated by our algorithm. Then our algorithm automatically samples the columns using the proper type according to the SQL operation. For example, the columns in the GROUP BY sub-clause will be only sampled from the categorical columns. Given a specific database, above efficient SQL generation method can automatically generate lots of SQL statements without any heavy human labor.

**Question Generation:** After generating SQLs on the specific database, we automatically transform SQLs into the corresponding natural language (NL) questions using the state-of-the-art LLM—GPT-4[6]. Recent research (Zhang et al., 2024a) benchmarked the capabilities of LLMs in various Text-to-SQL sub-tasks. Notably, GPT4 achieves the highest performance in the SQL-to-Text sub-task. It shows the potential of GPT-4 in automatically generating NL questions for given SQLs. However, directly prompting GPT-4 to translate complex SQL statements into NL questions always leads to inaccuracies, because GPT-4 may misunderstand nested structures and aliases within SQL statements or overlook critical details such as specific fields or filter conditions. To address these, we introduce a SQL decomposition step before translation, as referenced in the previous data augmentation work (Wu et al., 2021). Our algorithm first decomposes a SQL statement into several sub-clauses separated by SQL keywords such as SELECT, WHERE, GROUP BY, ORDER BY, etc. For nested structures in a SQL query, our algorithm recursively decomposes sub-queries and integrates them into a list of SQL clauses. We utilize GPT-4 with the few-shot prompting technique (Brown et al., 2020) to roughly translate SQL clauses into NL descriptions based on their functionalities and the information retrieved. Few-shot examples are randomly sampled from the training data for the clause-to-subquestion translation model (Wu et al., 2021). These NL descriptions for SQL clauses, referred to as hints, are fed into GPT-4 along with the database description. Finally, GPT-4 combines the original SQL statement, hints, and database description to produce the comprehensive translated NL question. The complete prompt is shown in Figure 8 in Appendix. With the hints and database descriptions, GPT-4 can generate more accurate and authentic questions, which are indistinguishable from those proposed by real users.

**Question-SQL Pair Selection:** Although our question-SQL pair generation algorithm incorporates well-designed techniques and strategies to improve semantic correctness, there may still be some abnormalities in the generated data, such as unreasonable column sampling, question-SQL inconsistency, and meaningless query results. To further enhance the quality and availability of the generated data, we define a new question-SQL pair selection step after generation. Recent research (Lin & Chen, 2023) on LLM evaluation for open-domain conversations demonstrates the effectiveness and efficiency of using LLMs to evaluate conversation quality. This research highlights the potential of LLMs as effective automatic evaluators. Therefore, we employ the reasoning capabilities of GPT-4 to evaluate and filter generated question-SQL pairs. We formulate four fine-grained scoring metrics to measure the quality of question-SQL pairs: *Question quality, SQL quality, Consistency* and *Significance*. 1) *Question quality* score reflects the clarity and fluency of the question and how relevant it is to potential users. 2) *SQL quality* score reflects the correctness of the SQL statement in terms of syntax and its ability to retrieve the correct data. The other two scoring metrics are assessed by considering the question-SQL pair as a unified entity. 3) *Consistency* score reflects the degree to which the SQL statement aligns with the intention of the question, aiding in the elimination of question-SQL inconsistency. 4) *Significance* score reflects the likelihood that the query would be posed by real users, as well as the informativeness and meaningfulness of the query results. It is designed to eliminate correct but meaningless question-SQL pairs. All scoring metrics are individually rated by GPT-4, with each sub-score ranging from 0 to 100. Finally, we retain

---

[6]https://openai.com/gpt-4

question-SQL pairs with all sub-scores greater than or equal to 80 as a high-quality, automatically generated dataset for constructing our benchmark. The prompt for question-SQL pair selection is composed of three parts: *criteria_prompt*, *one_shot_prompt* and *rate_qa_prompt*. *Criteria_prompt* outlines the specific scoring criteria, including *question_quality*, *SQL_quality*, *consistency*, *significance*. *One_shot_prompt*, containing a sample question-SQL pair and the corresponding scores, is used to standardize the output format of the LLM. *Rate_qa_prompt* involves question-SQL pairs to be scored and scoring instructions. The final prompt for the LLM input is shown in the Figure 6 in Appendix.

**Multi-turn Q&A Generation:** For the generation of multi-turn dialogues, we refer to the previous multi-turn data construction methods (Yu et al., 2019a;b). Differently from before, GPT-4 is used to automatically generate samples encompassing multiple Q&A types. We randomly combine single-turn Q&A types and utilize the chain-of-thoughts technique (Wei et al., 2024) to gradually expand single-turn Q&A. Validation and correction by LLM are used to improve the quality of generated Q&A pairs. Figure 4 in Appendix F shows the detailed generation pipeline.

**Other Metric Datasets Generation:** 1) For the generation of question-SQL pairs with external knowledge, we manually collect relevant external knowledge about the specific database according to the fine-grained categories. We then utilize GPT-4, combined with these external knowledge and database information, to generate question-SQL pairs involving external knowledge. 2) For the generation of question-SQL pairs containing multiple logical reasoning steps, we manually summarize several logical reasoning types, and then use GPT-4 to generate question-SQL pairs containing at least three logical reasoning steps. 3) For the generation of question-SQL pairs covering diverse NL questions, we utilize GPT-4 to rewrite user questions according to the types of NL questions defined manually. 4) For the generation of question-SQL pairs supporting multiple SQL dialects, we analyze and collect 21 unique characteristics for on SQL functions for five prevalent SQL dialects, including SQL Standard, PostgreSQL, SQL Server, MySQl and Oracle. For each SQL function in Table 8 in Appendix, we list the corresponding unique SQL keywords or functions for each SQL dialect, where the symbol "–"indicates no corresponding keywords. Based on functions mentioned in Table 8, we generate a total of 177 question-SQL pairs, covering these functions across four database systems.

Table 2: An overview comparison between our benchmark and other text-to-SQL benchmarks.

| Dataset | # Example | # DB | NL Diversity | Ext Knowledge | Logical Reasoning | Robustness | Interactivity | Generalizaton |
|---|---|---|---|---|---|---|---|---|
| WikiSQL (Zhong et al., 2017) | 80,654 | 26,521 | ✗ | ✗ | ✗ | ✗ | ✗ | ✗ |
| Spider (Yu et al., 2018) | 10,181 | 200 | ✗ | ✗ | ✗ | ✗ | ✗ | ✗ |
| KaggleDBQA (Lee et al., 2021) | 272 | 8 | ✗ | ✗ | ✗ | ✗ | ✗ | ✗ |
| BIRD (Li et al., 2024b) | 12,751 | 95 | ✗ | ✓ | ✗ | ✗ | ✗ | ✗ |
| ScienceBenchmark (Zhang et al., 2024b) | 5032 | 3 | ✗ | ✗ | ✗ | ✗ | ✗ | ✗ |
| SParC (Yu et al., 2019b) | 4,298 | 200 | ✗ | ✗ | ✗ | ✗ | 4,298 | ✗ |
| CoSQL (Yu et al., 2019a) | 3,007 | 200 | ✓ | ✗ | ✗ | ✗ | 3,007 | ✗ |
| ADVETA (Pi et al., 2022) | 11,455 | 178(+283 tables) | ✗ | ✗ | ✗ | ✓(Table Perturbation) | ✗ | ✗ |
| Dr. Spider (Chang et al., 2023) | 14999 | 200 | ✗ | ✗ | ✗ | ✓ | ✗ | ✗ |
| **Our benchmark** | 10,885 | 74 | 540 | 173 | 31 | 400 | 10,874 | 177 |

### 4.3 DATA STATISTICS

Our benchmark consists of 10,885 complex question-SQL pairs and 10,874 multi-turn dialogues over 74 public databases, covering 9 first-level metrics and 18 second-level metrics. In detail, our benchmark contains 9,964 question-SQL pairs automatically generated by our generation pipeline, 173 question-SQL pairs with external knowledge, 177 question-SQL pairs for testing generalization of SQL dialects, 31 question-SQL pairs with complex logical reasoning steps and 540 question-SQL pairs for NL question diversity. The databases collected in our benchmark spans more than 12 professional domains, including sports, finance, entertainment, health, science, etc. The detailed distribution of the databases is illustrated in the Figure 2. The 74 databases along with question-SQL pairs are divided into 3 subsets, 50 databases used as the training set, 13 databases belonged to the dev set, and 11 databases belonged to the test set. Among them, question-SQL pairs related to external knowledge, complex logical reasoning, NL question diversity, SQL dialect diversity all belong to the test set. Table 2 provides a statistical comparison between our benchmark and previously well-known benchmarks. The depth and width distribution of the SQLs in our benchmark compared to SPIDER and BIRD is depicted in Figure 5 in Appendix. Our benchmark includes more complex SQL queries with greater depth, and the width of our SQL queries reaches up to

80, significantly surpassing other benchmarks. The distribution of SQL keywords and functions is shown in Figure 3 in Appendix.

### 4.4 COST ANALYSIS OF DATA GENERATION

In our data generation pipeline, only the question generation and question-SQL pair selection parts use the OpenAI gpt-4-turbo API, incurring API usage fees. Though batching multiple SQLs or question-SQL pairs into one prompt, generating a question and scoring a question-SQL pair cost 0.01$ and 0.006$ respectively on average. Therefore, the cost of our data generation pipeline is only 0.016$ for each question-SQL pair on average, which is much lower than the cost of manual generation with intensive human labor (1.6$ for each question-SQL pair in BIRD (Li et al., 2024b)).

## 5 EXPERIMENTS

### 5.1 BASELINE MODELS

Incorporating LLMs with in-context learning (Dong et al., 2023) techniques is currently a popular approach for text-to-SQL implementation. We selected several recent state-of-the-art LLMs for general or specific domains as our baseline models to perform experiments on our benchmark, including GPT-3.5[7], GPT-4[8], Llama3[9], Code Llama (Rozière et al., 2024), Gemma[10], Mixtral[11], Qwen (Bai et al., 2023), WizardLM (Xu et al., 2024), and SQLcoder[12]. GPT-4 and GPT-3.5, developed by OpenAI, are advanced closed-source general natural language processing models. GPT-4 demonstrates significant improvements over GPT-3.5 in terms of linguistic accuracy, contextual comprehension, and task execution capabilities. Llama3, Gemma, Mixtral and Qwen are well-known open-source LLMs. WizardLM is fine-tuned Llama model based on generated complex instruction data. Code Llama is the code-specialized fine-tuned LLM based on Llama 2, which are expert in code generation. SQLCoder, developed by Defog AI, is a state-of-the-art specialized large language model for text-to-SQL tasks. Our experiments are conducted on a server with an Intel(R) Xeon(R) Gold 6133 CPU @2.50GHz and two NVIDIA A800 80GB PCIe GPUs utilizing a open-source LLM cloud platform[13].

### 5.2 EVALUATION METRICS

Following previous works (Li et al., 2024b; Yu et al., 2018; 2019a), we consider execution accuracy as our primary evaluation metric to measure the correctness of text-to-SQL systems. Following previous work (Yu et al., 2019b), we introduce question match and interaction match metrics to evaluate the interactivity of text-to-SQL systems. **Execution accuracy (EX)**: The EX Score is computed as the proportion of samples in the evaluation set where the execution results of the predicted SQL queries match those of ground-truth SQL queries. **Question Match (QM) and Interaction Match (IM)**: QM is the exact set matching score over all questions, and IM is the exact set matching score over all interactions. The exact set matching score is 1 for each question only if all predicted SQL clauses are correct, and 1 for each interaction only if there is an exact set match for every question in the interaction.

### 5.3 EXPERIMENTAL SETTINGS

We conduct our experiments on the LLM API platforms *together.ai* & *OpenAI* and a server with two NVIDIA A800 80GB PCIe GPUs. GPT-3.5, GPT-4, Llama3, Code Llama, Gemma, Mixtral, Qwen, and WizardLM are deployed on the LLM API platforms, while SQLcoder is deployed on our server. The ICL prompts input into LLMs for SQL generation are shown in Figure 7 in Appendix.

---

[7]https://platform.openai.com/docs/models/gpt-3-5-turbo

[8]https://openai.com/index/gpt-4

[9]https://llama.meta.com/llama3

[10]https://ai.google.dev/gemma

[11]https://mistral.ai/news/mixtral-of-experts

[12]https://github.com/defog-ai/sqlcoder

[13]https://www.together.ai

Table 3: Model EX accuracy performance comparison.

| Models | Dev Set | Test Set | | | | |
|---|---|---|---|---|---|---|
| | | SQL complexity | External knowledge | | Logical reasoning | NL diversity |
| | | | w/o knowledge | w/ knowledge | | |
| GPT-3.5 | 54.46 | 56.22 | 11.50 | 12.00 | 9.67 | 37.00 |
| GPT-4 | 62.16 | 62.13 | **27.00** | **28.00** | **16.12** | 40.00 |
| Llama3 (70B) | 49.60 | 57.72 | 12.71 | 15.60 | 12.90 | **40.55** |
| Gemma (7B) | 14.32 | 26.28 | 2.81 | 7.04 | 0.00 | 9.62 |
| Code Llama (7B) | 10.31 | 12.58 | 2.90 | 5.20 | 0.00 | 15.18 |
| Code Llama (34B) | 47.15 | 53.03 | 4.60 | 7.50 | 6.45 | 20.18 |
| Code Llama (70B) | 51.80 | 48.08 | 0.00 | 8.10 | 6.45 | 10.74 |
| SQLcoder (7B) | 6.06 | 5.95 | 0.00 | 0.00 | 0.00 | 2.03 |
| SQLcoder (34B) | 19.01 | 16.13 | 1.20 | 1.20 | 0.00 | 3.33 |
| SQLcoder (70B) | 23.59 | 21.03 | 1.20 | 1.20 | 3.22 | 4.07 |
| Mixtral (8x7B) | 33.98 | 46.91 | 2.90 | 12.70 | 0.00 | 20.18 |
| Qwen1.5 (7B) | 19.33 | 38.48 | 2.30 | 9.20 | 3.22 | 38.70 |
| Qwen1.5 (72B) | 42.89 | 57.27 | 12.10 | 17.90 | 12.90 | 38.70 |
| WizardLM (13B) | 35.49 | 32.97 | 0.58 | 2.30 | 0.00 | 4.25 |
| Code Llama-SFT (34B) | **74.67** | **74.57** | 6.93 | 10.98 | 3.22 | **40.55** |

For the sake of saving the number of tokens consumed, we batch ten questions into one prompt for generate SQLs. When generating SQLs with LLMs, we set $temperature = 0.7, top\_p = 1$. For fine-tuning Code Llama model with the training set for single-turn question-SQL pairs on our server, we use the LoRA fine-tuning method and set $r = 16$, $alpha = 32$, $lr = 5e - 5$, $batch\_size = 4$, $train\_epochs = 4$. For fine-tuning LLM with multi-turn dialogues, we use QLoRA and set $r = 64$, $alpha = 16$, $lr = 5e - 5$.

## 5.4 ACCURACY ANALYSIS

We first analyze the accuracy performance of baseline models on the dev set and test set of our benchmark. The dev set and the SQL complexity part of test set contains 2,772 and 1,959 complex question-SQL pairs respectively. In the following experiments, we construct the input prompt for LLMs with the CREATE SQL of the database and a command for instructing LLMs to generate SQLs. The results of model accuracy performance are shown in Table 3. Experimental results demonstrate that our benchmark is challenging to most advanced LLMs, and even the state-of-art model, GPT-4, only achieves about 62% EX accuracy. Although the open-source models including Code Llama and SQLcoder have been trained in previous benchmarks, their performance is inferior to that of the closed-source model due to the small scale of parameters and the lack of training on our benchmark. This indicates that our benchmark has made up for the missing data patterns in the previous benchmark. Code Llama SFT (34B) training on our train set can achieve nearly 75% execution accuracy, surpassing that of GPT-4. We then evaluate the accuracy performance of models on other metrics. Based on the *home-credit-default* financial database, the external knowledge, logical reasoning and NL question diversity parts of the test set contains 17,331 and 540 question-SQL pairs respectively. The results show that all models exhibit inadequate performance in handling these complex aspects, indicating significant room for improvement. It is essential to design targeted promotion strategies to address these deficiencies.

## 5.5 ROBUSTNESS ANALYSIS

To evaluate the robustness of baseline models, we construct 400 unanswerable questions classified in four types: everyday questions, ambiguous questions, outside-database questions, and Non-SQL questions based on the databases of test set. Each type contains 100 different questions. We prompt the baseline models to evaluate whether the current question can be answered using SQL. If the

Table 4: Comprehensive model performance analysis.

(a) Robustness performance

| Models | Everyday | Ambiguous | Outside | Non-SQL |
|---|---|---|---|---|
| GPT-3.5 | **100** | 0 | **100** | **100** |
| GPT-4 | **100** | 0 | **100** | 0 |
| Code Llama | 0 | 0 | 0 | 0 |
| SQLcoder | **100** | 70 | 40 | 70 |

(c) Interactivity performance

| Models | QM | IM |
|---|---|---|
| GPT-3.5 | 32.20 | 10.40 |
| GPT-4 | 33.60 | 10.80 |
| Code Llama | 6.40 | 1.40 |
| Code Llama-SFT | **50.90** | **24.20** |
| SQLcoder | 0.70 | 0.00 |
| SQLcoder-SFT | 46.20 | 19.80 |

(b) Model's support for SQL dialect features

| Models | PostgreSQL | SQL Server | MySQL | ORACLE | Overall |
|---|---|---|---|---|---|
| GPT-4 | **44.44** | **38.89** | 77.78 | **57.89** | **54.79** |
| GPT-3.5 | 10.00 | 0.00 | 63.64 | 36.36 | 16.43 |
| Code Llama | 33.33 | 0.00 | **83.33** | 26.32 | 35.61 |
| SQLcoder | 38.89 | 33.33 | 33.33 | 31.58 | 34.24 |

question cannot be answered based on the database information, the models output *cannot answer*. We investigate the baseline model's responses to confusion questions. As shown in Table 4a, the number indicates the ratio of correctly identified unanswerable questions. The results show that none of the baseline models can accurately handle all types of confusion questions. Ambiguous questions are the most confusing and challenging for all models. It is worth noting that GPT-4 outputs SQLs when encountering ambiguous and Non-SQL questions, but GPT-4 will output extra explanation about the ambiguity of the questions or output SQLs to extract required information for the next Non-SQL operations.

## 5.6 INTERACTIVITY ANALYSIS

To evaluate the interactivity of baseline models and the quality of our generated data, we fine-tune the open-source LLMs on our multi-turn dialogues, and then test the interactivity performance of models on CoSQL (Yu et al., 2019a) dev set. The QM and IM results are shown in Table 4c, which show that untrained open-source models cannot process other types of questions in multi-turn dialogues. After training these models, there is an increase of nearly 50% in QM and 20% in IM. This demonstrates that our automatically generated data can effectively improve the interactivity performance of the model.

## 5.7 GENERALIZATION ANALYSIS

We summarized total 21 features for four SQL dialects, including PostgreSQL, SQL Server, MySQL and Oracle. We construct specific question-SQL pairs for each feature and test 73 question-SQL pairs on baseline models. The comparison of baseline models on generalization ability is shown in Table 4b. GPT-4 supports the most features across four SQL dialects, showcasing its strong generalization ability. SQLcoder, which claims to support multiple database systems, also covers nearly half of the SQL dialect features. The experimental results on generalization analysis indicate that the current support for SQL dialects of current LLMs is not comprehensive enough and needs to be strengthened.

## 6 CONCLUSION

Our multidimensional text-to-SQL benchmark, OCTOPUS, comprising 10,885 complex question-SQL pairs and 10,874 multi-turn dialogues across 74 public databases. With well-designed metrics and automatic generation pipelines, OCTOPUS not only provides a more fine-grained evaluation of text-to-SQL model performance, but also reveals deficiencies in current text-to-SQL models. Depending on our automatic data generation pipeline, OCTOPUS is highly scalable and can be expanded at a low cost. The evaluation of advanced text-to-SQL models on OCTOPUS has revealed considerable room for improvement, highlighting the necessity for further research and development in this domain.

## 7 ETHICS STATEMENT

The development of the OCTOPUS benchmark was guided by ethical considerations regarding data sourcing, model usage, and the potential impact of the research. All databases utilized in our benchmark were sourced from publicly available and open-source repositories, and were carefully selected to ensure they do not contain personally identifiable information (PII) or other sensitive data. We acknowledge that the large language models (LLMs) used for data generation, such as GPT-4, may reflect biases present in their training corpora. However, the primary purpose of our work is to create a comprehensive evaluation tool that can help researchers identify and ultimately mitigate such weaknesses in text-to-SQL systems. The overarching goal of this research is to advance technology that democratizes data access for non-technical users, which we believe is a positive societal contribution. Furthermore, our automated generation pipeline is designed to reduce the reliance on intensive manual annotation, promoting a more sustainable and cost-effective approach to benchmark creation.

## 8 REPRODUCIBILITY STATEMENT

We are committed to ensuring the reproducibility of our research. To this end, we have provided detailed descriptions of our experimental setup in Section 5, including the baseline models (Section 5.1), evaluation metrics, and implementation settings. Crucially, the full prompts used for data generation and model evaluation are included in the Appendix (Figures 6, 8, and 7) to allow for precise replication of our interactions with the large language models. Upon publication of this paper, we will publicly release the complete OCTOPUS benchmark dataset, including all question-SQL pairs, multi-turn dialogues, and associated database schemas. Furthermore, we will release all the source code used for data generation, model evaluation, and analysis. This will enable other researchers to verify our results, build upon our work, and use OCTOPUS to evaluate their own text-to-SQL systems.

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

# APPENDIX

## A    THE DETAILED DEFINITIONS OF FIRST-LEVEL AND SECOND-LEVEL METRICS

The detailed definitions of the first-level and second-level metrics not mentioned in the main paper are illustrated below.

Table 5: Definition of external knowledge categories.

| Primary Category | Secondary Category | Definition |
| --- | --- | --- |
| Database Internal Knowledge | Table Definition | Descriptions of the meaning of an entire table in the database |
| | Field Description | Descriptions of the meanings of field names in a table |
| | Value Description | Detailed descriptions of database values, including value types, ranges, and categories |
| Database External Knowledge | Concept Description | Detailed explanations of relevant noun concepts in a specific domain |
| | Calculation Description | Calculations and formulas associated with database fields |
| | Relation Description | Entity relationship descriptions for fields and values, such as inclusion and composition |
| | Constant Definition | Constants and statistical data values related to specific domains |
| | Abbreviation/Alias | Descriptions of abbreviations and aliases for specific domains |

**Database Complexity.**    We focus on the complexity of the data model when considering the concept of database complexity. A data model with lengthy field names and complex foreign key associations will increase the difficulty of extracting relevant table information and generating JOIN clauses for text-to-SQL system based on user questions. In order to test the understanding and information extraction capabilities of a text-to-SQL system on complex database structures, we further decompose database complexity metric into three second-level metrics, containing ***field naming complexity, table similarity***, and ***table coupling degree*** from perspective of increasing the difficulty of SQL generation.

- ***Field Naming Complexity.*** This metric requires the benchmark dataset to contain diverse and complex field naming method. We expect the benchmark to contain databases with different field naming styles including *English, Chinese*, and *Abbreviation*. Meanwhile,

we expect the database field names to be as long as possible to test if the LLM based on probability can output the field name accurately and stably.

- **Table Similarity.** Table similarity is defined as the overlapping degree of the set of fields of two tables. The higher the overlapping degree of two tables, the closer their semantic meanings are and the harder for LLMs to distinguish them. This metric requires the databases in our benchmark to contain tables with high similarity as many as possible. It is designed to examine the text-to-SQL system's ability of retrieval for relevant tables and columns.

- **Table Coupling Degree.** We defined the table coupling degree of a database as the density of database graph where tables are treated as nodes and foreign key constraints are regarded as edges. Higher table coupling degree means that querying the same information needs more JOIN operations in one SQL, which is a key challenge in the text-to-SQL translation process. This metric requires the benchmark to contain the databases with higher table coupling degree to inspect the ability of JOIN clause generation for complex SQL.

**Gold SQL Complexity.** Gold SQLs are correct SQL statements corresponding to user questions, which should be predicted by text-to-SQL models. We measure the gold SQL complexity from the following two perspectives: **SQL structural complexity** and **SQL operation diversity**. We design this metric to concentrate on assessment of text-to-SQL system's ability for SQL generation part.

- **SQL structural complexity**. We utilize the depth and width of the AST(Abstract Syntax Tree) of SQL statements to assess the SQL structural complexity. A deeper AST structure often indicates that the SQL statement has a higher level of nested structure, meanwhile, a wider AST structure implies that the SQL statement has a greater number of clauses, which poses a great challenge for SQL generation. We set this metric to guide our benchmark to contain more deeper and wider SQL samples to test text-to-SQL system's ability in aspect of generating complex SQLs.

- **SQL operation diversity**. To enhance the comprehensiveness of our benchmark, it is imperative to include SQL samples that not only exhibit more intricate structures, but also encompass a broader array of SQL operations, including diverse keywords, functions, and additional syntactic elements. We aim for our benchmark to encompass as wide a range of SQL syntactic operations as possible to assess the text-to-SQL model's comprehension of SQL syntax.

**Natural Language(NL) Question Diversity.** Natural language questions are proposed by users and treated as input by text-to-SQL systems. Most text-to-SQL systems do not format questions input by users, thus natural language questions containing different kinds of forms and variations. Taking the impact of natural language questions on accuracy for text-to-SQL systems into consideration, we depict the natural language question diversity into two fine-grained metrics: **Diversity in NL Questioning Ways** and **Ambiguity of NL Questions**.

- **Diversity in NL Questioning Ways**. This second-level metric concentrates on the different ways of natural language questions being asked regardless of information it contains. For example, we regard *"What are the maximum and minimum budget of the departments?"* and *"List the maximum and minimum budget of the departments?"* as two different questioning ways but querying the same information. In our benchmark, we change the questioning ways of users' queries to test if the accuracy of text-to-SQL system will decline under different circumstances.

- **Ambiguity of NL Questions**. Due to the fact that users of text-to-SQL systems are usually not expert in database systems and SQL syntax, questions proposed by them could be ambiguous and vague, including synonyms and implications. This metric is designed to simulate ambiguous natural language questions by categorizing different types of such questions as encountered in real-world scenarios, in order to evaluate the capability of text-to-SQL systems to respond accurately to these questions.

**Logical Reasoning Complexity.** A complex SQL statement frequently involves intricate logical reasoning steps, which can be decomposed into a sequence of simpler SQL query statements in

order. We define a logical clause in a complex SQL query, such as JOIN, GROUP BY, SORT, or SUBQUERY clauses, which can be dismantled to obtain an intermediate result, as a constituent step in the logical reasoning process. Logical reasoning complexity means the number of logical reasoning steps required to generate one complex SQL statement. Our benchmark introduces this metric and constructs question-SQL pairs with high logical reasoning complexity to test the reasoning capabilities of text-to-SQL systems.

**External Knowledge Complexity.** Real-world databases cover multiple domains and different definitions. Incorporating descriptions of databases and domain-specific knowledge is essential for text-to-SQL systems to generate executable and accurate SQL queries. For example, considering a situation where a financial database contains one table *Transactions(ID, Revenue, Cost, Date)* and a user question is *"Calculate the total profit for all transactions."*, the text-to-SQL system must comprehend *the calculation of profit* corresponding to the equation *Profit=Revenue-Cost* to generate the correct SQL *"SELECT SUM(Revenue - Cost) AS TotalProfit FROM Transactions"*. Based on the analysis of user problems that may occur in real scenarios and the integration of other related research on external knowledge, we summarize and classify the external knowledge that text-to-sql system may need to generate correct SQLs, as detailed in Table 5. In our paper, we consider that the external knowledge can be divided into two main categories based on the relevance to the database, one is the internal knowledge of the database which is database-specific, the other is knowledge external to the database the database which contains common sense and domain-specific knowledge. We further split these two kinds into more fine-grained categories. We set the variety and number of external knowledge required for generating gold SQLs as the second-level metrics to form a subset dataset containing various user questions related to external knowledge for testing the text-to-SQL systems' ability to retrive and understand external knowledge.

**Confusion Question Testing.** In real-world scenarios, user questions are diverse and may include some distracting questions. We have defined four types of confusion questions referring to the Dr.spider benchmark (Chang et al., 2023): *Everyday Conversations (Non-SQL Q&A), Ambiguous Questions (one question corresponding two or more correct SQLs), Unanswerable Questions(querying information outside the database)*, and *Unsupported Questions(querying operations not supported by SQL statements such as plotting figures)*. We introduce these confusing questions to test the ability of the text-to-SQL system to identify and process questions that cannot be answered.

**Perturbation Testing.** A robust text-to-SQL system must exhibit resilience against potential variations that may arise during the course of user interaction. To evaluate the robustness of the text-to-SQL systems, we stimulate two real-world common kinds of perturbations in our benchmark: *Perturbations to the database* and *Perturbations to natural language questions*. It is common for database tables, fields, and records to be updated in enterprise applications, and user questions may also contain replacement changes. We aim to simulate these two types of perturbation through constructing samples to measure how robust a text-to-SQL system is.

## A.1 DEFINITION OF EXTERNAL KNOWLEDGE CATEGORIES

We further divide external knowledge into two categories: **Database Internal Knowledge** and **Database External Knowledge**.

Database internal knowledge refers to specific information about databases that cannot be accessed or obtained through a standard connection to the database system. In real applications, database administrators often use abbreviations to name tables and columns (also called fields) for convenience. These tables and fields are frequently not well-documented within the database, making it challenging for text-to-SQL systems to comprehend the database structure. Although we can obtain the data types and sample values of fields through the database system, the specific meanings represented by these values (e.g., 'F' representing 'female') are difficult for text-to-SQL systems to understand. Therefore, text-to-SQL systems need to combine detailed descriptions for the specific database including **table definition**, **field description** and **value description** as external knowledge to generate the correct SQL statement. This kind of external knowledge varies with different databases.

Database external knowledge refers to the external information independent of the specific database, which is often common sense or domain knowledge. We further divide database external knowledge into 5 secondary categories: **Concept Description**, **Calculation Description**, **Relation Description**, **Constant Definition** and **Abbreviation/Alias**. The definitions of these categories are listed in the Table 5. Let us give a concrete example for each category to make it easier to understand. For example, DPD, short for Days Past Due (*Abbreviation/Alias*), indicates how many days have passed since the due date of the loan or credit card payment (*Concept Description*). Profit equals revenue minus cost, which is a simple example of *calculation description*. *Relation description* involves inclusive and non-inclusive relationships, such as "China belongs to an Asian country". *Constant definition* describes the specific values not included in the database corresponding to the concept. $\pi = 3.1415926$ is a simple example for this category. In our samples related to external knowledge in the dataset, we have detailed annotations of the involved external knowledge descriptions and classifications. It is worth noting that a single sample may involve multiple entries of external knowledge.

## A.2 THE DEFINITIONS OF CATEGORIES FOR NL DIVERSITY

We summarize and define the types of possible variations on user questions in practical text-to-SQL systems in Table 7. These types are used to guide the generation of test samples for NL diversity. We paraphrase the original questions of question-SQL pairs to rewritten questions with the help of GPT-4. A rewritten question can involve multiple types of NL variations. All rewritten questions are annotated with types and descriptions of each included variation .

## B THE COLLECTION OF UNIQUE CHARACTERISTICS FOR DIFFERENT SQL DIALECTS

We analyze and collect 21 unique characteristics for on SQL functions for five prevalent SQL dialects, including SQL Standard, PostgreSQL, SQL Server, MySQl and Oracle. For each SQL function in Table 8, we list the corresponding unique SQL keywords or functions for each SQL dialect, where the symbol "–"indicates no corresponding keywords. Based on functions mentioned in Table 8, we generate a total of 177 question-SQL pairs, covering these functions across four database systems.

## C THE COST ANALYSIS OF DATA GENERATION PIPELINE

In our data generation pipeline, only the question generation and question-SQL pair selection parts use the OpenAI gpt-4-turbo API, incurring API usage fees. Though batching multiple SQLs or question-SQL pairs into one prompt, generating a question and scoring a question-SQL pair cost 0.01$ and 0.006$ respectively on average. Therefore, the cost of our data generation pipeline is only 0.016$ for each question-SQL pair on average, which is much lower than the cost of manual generation with intensive human labor.

## D THE STATISTICS OF THE DISTRIBUTION OF SQL KEYWORDS AND FUNCTIONS

We conduct a statistical analysis of the keywords and functions in the SQL statements we generated. We create the following visual word cloud in Figure 3 based on the frequency of keywords and functions. We categorize all the keywords and functions into the following eight categories:

**Basic comparison and logical operators.** $=, \neq, >, <, \geq, \leq, <>, ||, +, -, \times, /$, OR, AND, IS, NOT

**Data aggregation functions.** COUNT(), AVG(), SUM(), MIN(), MAX(), GROUP_CONCAT(), GREATEST()

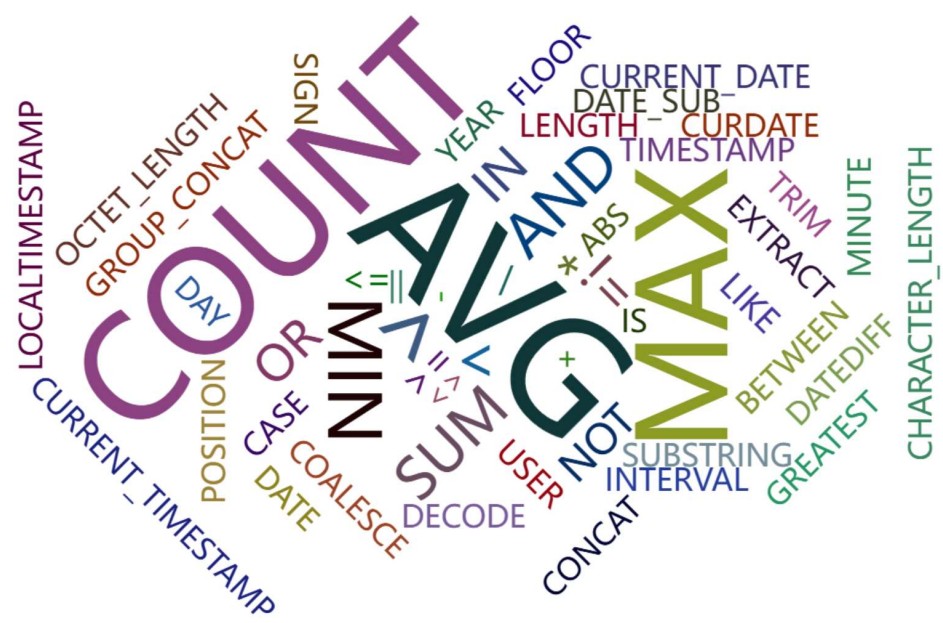

Figure 3: SQL keyword and function distribution

**Date and time handling functions.** TIMESTAMP(), MINUTE(), DATE_SUB(), EXTRACT(), DAY(), DATEDIFF(), DATE(), CURRENT_TIMESTAMP, CURRENT_DATE, CURDATE(), INTERVAL(), YEAR(), LOCALTIMESTAMP

**String handling functions and pattern matching.** LIKE, POSITION(), LENGTH(), CONCAT(), TRIM(), SUBSTRING(), CHARACTER_LENGTH(), OCTET_LENGTH()

**Conditional statements and data type handling.** IN(), COALESCE(), CASE, BETWEEN

**Mathematical operators and functions.** FLOOR(), ABS(), SIGN()

**Encryption and encoding functions.** DECODE()

**Other miscellaneous functions.** USER()

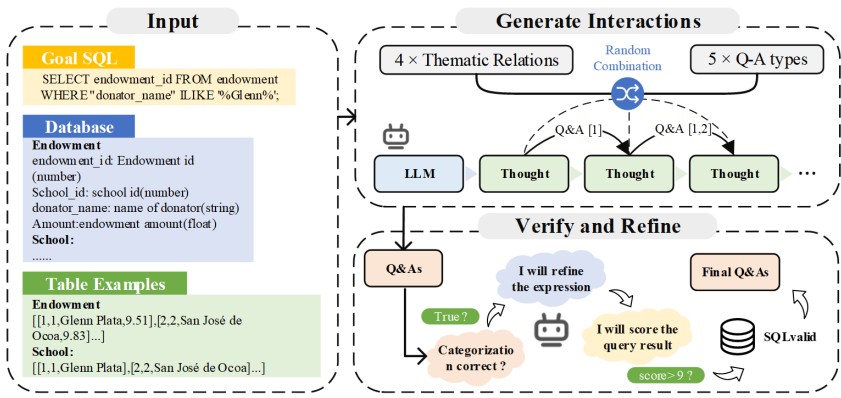

Figure 4: Overview of the pipeline for automatic multi-turn dialogues generation

# E  THE DETAIL OF PROMPTS

## E.1  PROMPT FOR QUESTION-SQL PAIR SELECTION

The prompt for question-SQL pair selection is composed of three parts: *criteria_prompt*, *one_shot_prompt* and *rate_qa_prompt*. *Criteria_prompt* outlines the specific scoring criteria, including *question_quality*, *SQL_quality*, *consistency*, *significance*. *One_shot_prompt*, containing a sample question-SQL pair and the corresponding scores, is used to standardize the output format of the LLM. *Rate_qa_prompt* involves question-SQL pairs to be scored and scoring instructions. The final prompt for the LLM input is shown in the Figure 6.

## E.2  QUESTION GENERATION PROMPT

Question generation for specific SQL consists of three steps: rough translation to convert SQL clauses into spoken English, clause translation to prepare hints and final translation which combines the results of rough translation and clause translation to obtain the final corresponding question. Figure 8 shows the detailed prompt for each translation step.

## E.3  PROMPT FOR TESTING SQL GENERATION

The prompt for testing SQL generation consists of three main components: *base_prompt*, *external_knowledge_prompt*, and *sql_dialect_prompt*. The *base_prompt* instructs the model to generate SQL queries based on provided SQL tables and user requests, returning the results as a list. The *external_knowledge_prompt* extends this by incorporating additional context from external knowledge sources to enhance the relevance of the generated queries. The *sql_dialect_prompt* further specifies that each generated SQL must conform to a particular SQL dialect, using predefined functions and database system types. The final prompt used as input to the LLM is illustrated in Figure 7.

# F  MULTI-TURN DIALOGUE GENERATION PIPELINE

Figure 4 shows the generation pipeline for multi-turn dialogues. The pipeline for automatic multi-turn dialogue generation starts with a target SQL query and a database schema, generating interactions through a large language model (LLM) using thematic relations and multiple Q&A types. These interactions are randomly combined to create diverse dialogue scenarios. The generated Q&As are then verified and refined by checking categorization, refining expressions, and scoring the results. Only those with high scores (e.g., score $> 9$) are deemed valid, resulting in a set of coherent and accurate multi-turn dialogues that reflect the original SQL queries and database schema.

# G  ERROR ANALYSIS ON TEST SAMPLES FOR LOGICAL REASONING

In our experimental results, we discover that all models perform poorly on the test set for logical reasoning, even the state-of-the-art GPT-4 model achieves only about 16% accuracy. The test set for logical reasoning is comprised of 31 complex question-SQL pairs which involves multiple difficult logical reasoning steps including sub-query, group-by, sort and join operations. Figure 9 shows an example of test set for logical reasoning, which needs three logical reasoning steps for LLM to correctly generate the gold SQL. It requires the text-to-SQL model to have a deep understanding of the database, stable SQL generation capabilities, and strong logical reasoning abilities. We analyze the results of LLMs in our experiments on these complex questions, and observe that the majority of errors are due to the execution of generated SQL statements, with logical errors accounting for only a minority. This indicates that the current text-to-SQL model has significant limitations in its capability to generate complex SQL statements that involve intricate logical reasoning steps.

# H  EVALUATION OF GENERATED DATA QUALITY

Due to the fact that the sampling rules for generating SQL cannot fully cover all potential cases that may lead to semantic issues, which may result in the generation of nonsensical SQL queries, we

Table 6: Correlation and Significance Levels

| Metric | Pearson | | Spearman | | Kendall | |
|---|---|---|---|---|---|---|
| | Correlation | P-value | Correlation | P-value | Correlation | P-value |
| question_quality | 0.561 | 1.18e-9 | 0.514 | 4.36e-8 | 0.424 | 3.55e-8 |
| SQL_quality | 0.873 | 1.90e-32 | 0.875 | 9.29e-33 | 0.778 | 5.30e-24 |
| consistency | 0.903 | 6.12e-38 | 0.851 | 3.01e-29 | 0.712 | 2.95e-22 |
| significance | 0.788 | 1.97e-22 | 0.770 | 6.72e-21 | 0.628 | 5.70e-15 |

adopted a strategy of using GPT-4 to evaluate and filter the generated question-answer pairs to ensure high-quality outputs. The idea of using large models for evaluation originates from previous work that employed large models to assess the quality of conversations and text-to-3D data, including G-EVAL (Liu et al., 2023), SummEval (Fabbri et al., 2021) and GPTEval3D (Wu et al., 2024), which is known as LLM-eval research filed. We follow the statistical tests in these papers to verify the alignment between GPT-4 and human experts on assessing generated text-to-SQL data. We collected 5 SQL experts to score one hundred randomly selected samples (including both high-quality and low quality data samples), according to the same scoring criteria provided to GPT-4. We introduce a cross-validation method in the scoring process. Each sample will be evaluated by two experts. If the score difference is less than 20, the average of the two scores will be taken as the final score. If the score difference exceeds 20, the experts will re-evaluate the sample through consultation until the score difference is reduced to less than 20. Finally, following the method of analysis used in previous work, we calculated Pearson, Spearman, and Kendall's Tau correlation coefficients along with their corresponding p-values for the scores given by GPT-4 and human experts. The final statistical analysis results are presented in Table 6. All the correlation coefficients and p-values mentioned above indicate that the human expert scores and GPT-4 scores have a strong positive correlation across all four scoring dimensions. This demonstrates the effectiveness of GPT-4's quality scoring and its consistency with human evaluations. We then analyzed the results of the high-quality and low-quality datasets selected by GPT-4 and human experts. Among these 100 random samples, 89% of the results from GPT-4 were consistent with those of human experts. Only 2% (2 out of 100) of the samples that humans deemed low quality were mistakenly identified as high quality by GPT-4. Additionally, 9% (9 out of 100) of the high-quality data samples were misclassified as low quality by GPT-4. This suggests that GPT-4 may apply stricter criteria than human experts. However, this does not compromise the overall quality of the final dataset.

## I  LICENSES FOR OPEN-SOURCE DATABASES

The databases in our benchmark are all in accordance with one of following licenses:

**Public Domain**  Public Domain Mark
A public domain license refers to a legal designation that allows intellectual property, such as creative works or inventions, to be freely used, shared, and built upon by anyone without restrictions. When a work is in the public domain, it is no longer protected by copyright, patent, or trademark laws.

**CC-BY**  Creative Commons Attribution 4.0 International
This license is one of the open Creative Commons licenses and allows users to share and adapt the dataset so long as they give credit to the creator.

**CC-BY-SA**  Creative Commons Attribution-ShareAlike 4.0 International
This license is one of the open Creative Commons licenses and allows users to share and adapt the dataset so long as they give credit to the creator and distribute any additions, transformations, or changes to the dataset under this license.

**GPL**  General Public License
The GPL was created by the Free Software Foundation (FSF) and is also known as the GNU GPL, as it is used by the GNU Project. And it allows users to use, study, share, and modify the software under certain terms and conditions.

**CPOL**  Code Project Open License
It is a software license that is often used for articles, tutorials, and sample code shared on The Code Project website. The CPOL is intended to be a more permissive license, allowing developers to use, modify, and distribute the software without many of the restrictions imposed by other licenses like the GPL.

**CC0**  Creative Commons Zero
It is a public domain dedication tool created by Creative Commons. It allows creators to waive all their copyright and related rights in a work, effectively placing it in the public domain. This means that anyone can freely use, share, modify, and build upon the work without seeking permission or providing attribution to the original creator.

## J  GENERATIVE AI USAGE STATEMENT

We utilized a large language model (LLM) to assist in the preparation of this paper. The LLM's role was strictly limited to improving grammar, clarity, fluency, and overall readability. It is important to distinguish this use from the application of LLMs as a core component of our research methodology. The utilization of GPT-4 for the automatic generation of questions and the filtering of question-SQL pairs, as part of the OCTOPUS benchmark creation pipeline, is a central aspect of our technical contribution and is described in detail in Section 4. We meticulously reviewed, revised, and edited all text to ensure it accurately reflects our research and findings. Full responsibility for the scientific content, claims, and final wording of this paper rests entirely with the human authors.

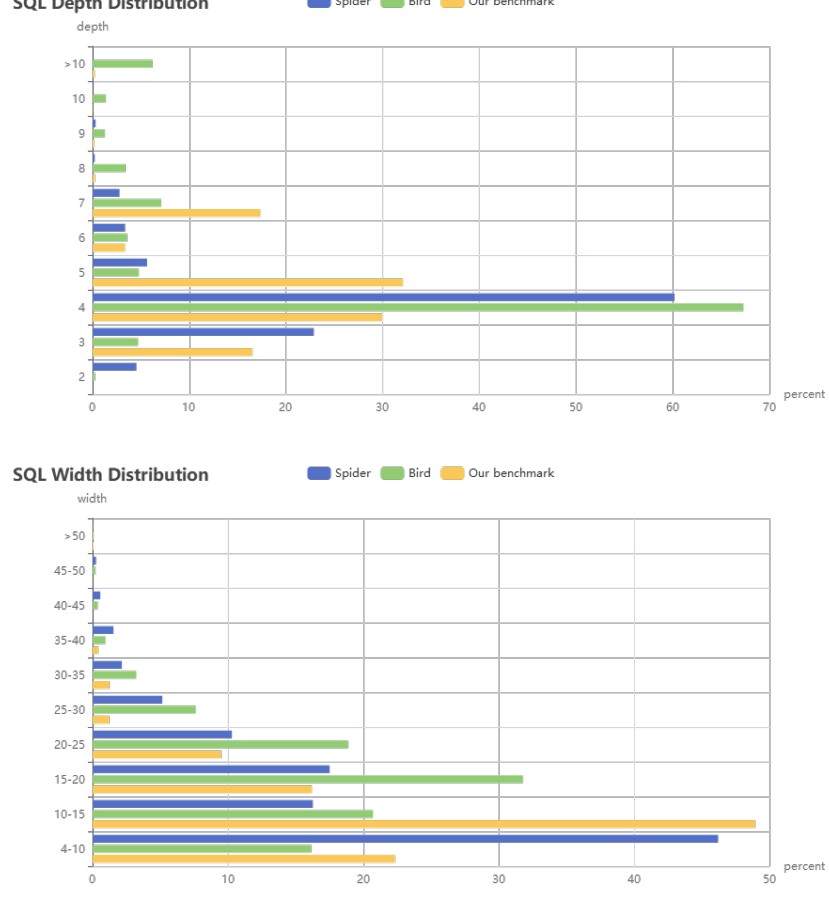

Figure 5: Depth and width distribution of SQLs.

Table 7: The definitions and examples of NL diversity categories.

| Type | Definition | Question | Paraphrases | Description |
|------|-----------|----------|-------------|-------------|
| **keyword synonym** | Use synonyms of keywords in SQL to rewrite the question. | Find the code of airport that has the highest number of flights. | Show me the code for the airport that currently has the most flights. | find and show are both synonyms for select |
| **keyword implicit** | Use the implicit expression of keywords in SQL to rewrite the question. | Arrange the test scores in descending order, who is ranked 5th? | Who is the student with fifth place in the exam? | fifth place implies order by desc |
| **operator synonym** | Use synonyms of operator or function in SQL to rewrite the question. | What is the code of airport that has the highest number of flights? | Show me the code for the airport that currently has the most flights. | the most is synonyms for max() |
| **operator implicit** | Use implicit expression of operator or function in SQL to rewrite the question. | Show the name and theme for all concerts and the number of singers in each concert. | List the names and themes for all concerts and how many singers are in each. | how many implicts count() |
| **column synonym** | Use synonyms for columns in database tables to rewrite the question. | List the name of teachers whose hometown is not Little Lever Urban District. | Find the name of teachers who were not born in Little Lever Urban District. | born in is synonym of hometown |
| **column implicit** | Use implicit expression of columns in database tables to rewrite the question. | Show the name of teachers aged either 32 or 33? | Which teachers are aged either 32 or 33. | which implicit name |
| **column attribute** | Use attributes of columns in database tables to rewrite the question. | What is the name of the conductor who has worked the greatest number of year? | Who has worked the longest as conductor? | longest represents an attribute of year |
| **column value** | Use value of columns in database tables to rewrite the question. | What are the ids of the students who do not own cats as pets? | Find the IDs of students who don't own cats. | cats is a value in the pets column |
| **column subset** | Use subset of columns in database tables to rewrite the question. | how many dogs are there? | how many puppies are there. | puppy refers to the column dog |
| **column shuffling** | Rewrite the question by shuffling the order of the columns in the returned database table | List students' names, grades, and classes. | List students' grades, names, and classes. | Shuffle the order in which columns are returned |
| **column abbreviation** | Use abbreviation of columns in database tables to rewrite the question. | What is China's GDP in 2023? | What is Gross Domestic Product of China in 2023 | find and show are both synonyms for select |
| **grammar conversion** | change the question by changing the sentence pattern, such as changing the question method, etc. | Find all technical department employees whose salary is higher than 5000. | Find all technical department employees with a salary higher than 5,000. | Two sentences have the same semantics but different grammatical structures |

Table 8: SQL function/method comparison across different database systems.

| Method/Function | SQL Standard | PostgreSQL | SQL Server | MySQL | Oracle |
|---|---|---|---|---|---|
| String Length | CHARACTER_LENGTH | CHARACTER_LENGTH CHAR_LENGTH LENGTH (all equivalent) | LEN | CHARACTER_LENGTH CHAR_LENGTH | CHARACTER_LENGTH CHAR_LENGTH LENGTH |
| Substring | SUBSTRING | SUBSTRING | SUBSTRING | SUBSTRING | SUBSTR |
| Trim Whitespace | TRIM | TRIM | LTRIM RTRIM | TRIM | TRIM |
| Local Timestamp | LOCALTIMESTAMP | LOCALTIMESTAMP CURRENT_TIME | CURRENT_TIMESTAMP GETDATE() SYSDATETIME | LOCALTIMESTAMP NOW CURTIME() | LOCALTIMESTAMP SYSTIMESTAMP |
| String Concatenation | string1 \| string2 | – | string1 + string2 | CONCAT() | string1 \| string2 |
| Null Handling | – | COALESCE | ISNULL | IFNULL | NVL |
| Date Truncation | – | DATE_TRUNC | N/A | DATEFORMAT | TRUNC |
| String Search | – | POSITION | CHARINDEX | LOCATE | INSTR |
| Time Difference | – | AGE+EXTRACT | DATEDIFF | DATEDIFF | +/- INTERVAL |
| Paging Query | – | LIMIT FETCH | FETCH | LIMIT FETCH | FETCH |
| Simple Conditional Control | – | – | IIF | – | DECODE |
| Flashback Query | – | – | – | – | TIME |
| Group Deduplication/Field Deduplication | – | DISTINCT ON | – | – | – |
| Sampling | – | TABLESAMPLE | TABLESAMPLE | ORDER BY RAND() | SAMPLE(PERCENTAGE) |
| Value Retrieval by Index | – | – | CHOOSE | ELT | – |
| String Regex Match and Split | – | REGEXP_SPLIT_TO_TABLE STRING_TO_ARRAY REGEXP_MATCHES (recommended) | STRING_SPLIT | REGEXP_SUBSTR | REGEXP_SUBSTR |
| String Byte Length | OCTET_LENGTH | OCTET_LENGTH | DATALENGTH | LENGTH | LENGTHB |
| Multi-group String Concatenation | – | STRING_AGG | STRING_AGG | GROUP_CONCAT | LISTAGG |
| Float Truncation | – | TRUNC | FLOOR | TRUNCATE | TRUNC |
| Array Merge/Collection Generation | – | – | – | – | COLLECT |
| Full Join | – | FULL JOIN | FULL JOIN | LEFT JOIN + UNION + RIGHT JOIN | FULL JOIN |

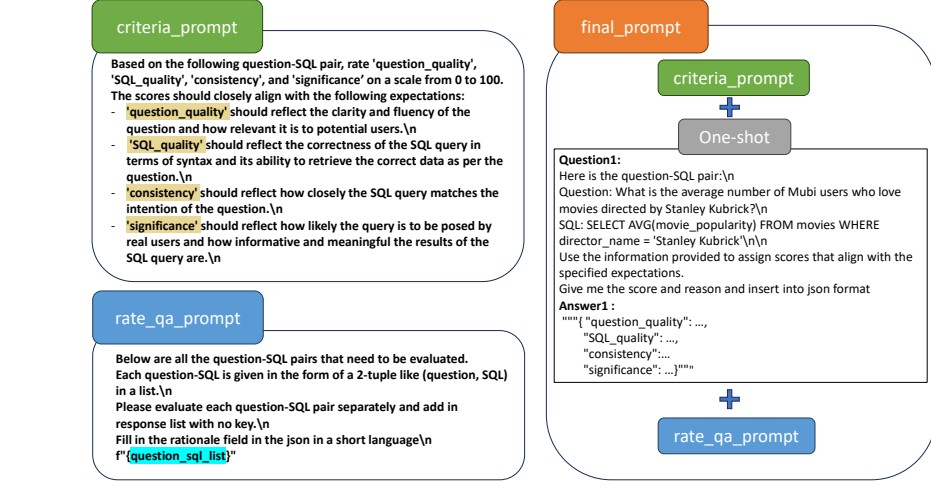

Figure 6: Prompt for question-SQL pair selection

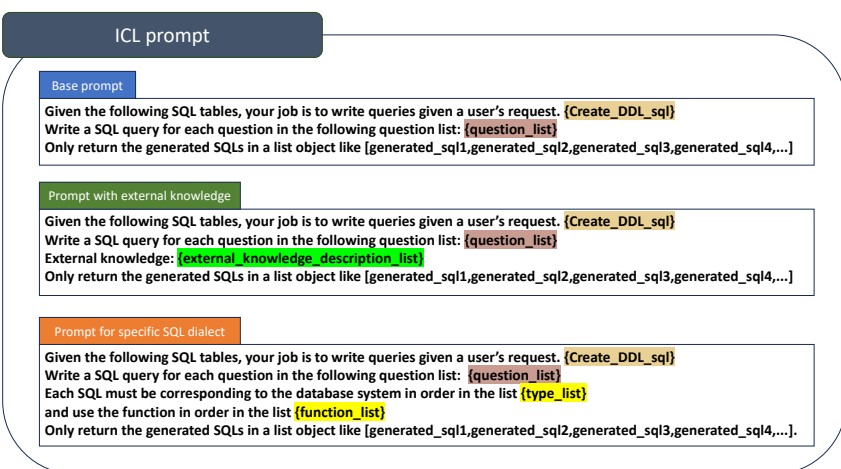

Figure 7: Prompt for SQL generation in experiments

## Rough translation prompt

Translate the sentences provided into spoken English, not contain any special symbols except commas and periods.
Only returns the translated sentences, do not generate any other content, such as "The clause is: ...".
If you have trouble doing it or hold that nothing need to be changed, just return the original sentences.
The returned sentences is either the translated sentences or the original sentences, but we encourage you to translate the sentences even if the changes are minor.
The sentences is: {sentences}
Make sure to return the result in JSON format: {"sentence1":"","sentence2":"", ...}

## Clause translation prompt with few-shot

You are a language expert. You need to translate the sql clauses provided and then return the colloquial result.
Here is some examples: {examples}
You need to translate the sql clauses provided and then return the colloquial result.
Do not generate any other content, such as "The paragraph means: ", just the result.
If you can't translate the sql clauses, return the clauses provided, don't reply content as such 'The SQL clauses is not valid. Please provide a valid SQL clauses.'.
The sql clauses is: {clauses}
Make sure to return the result in JSON format: {"sentence1":"","sentence2":"", ...}

## Final translation prompt

You are a sql to question translation expert. You need to translate the sql provided and then return the coresponding question easy to understand.
…
Here is some examples: {random_selection_examples}
You need to translate the sql provided and then return coresponding question.
Because sometimes SQL is too complex to understand, or even contains errors, we will also provide a simplified version(a clause list, contains main information about the sql) for reference.
In addition, we will also provide some tips to introduce the meaning of this sql, and what the generated question is mostly about.
…
We will provide the descriptions of the columns in SQL, which you can take as reference: {columns info}
The original sql is: {original_sql}
The simplified version is: {simplified_version}
The hints are: {hints}
Make sure to return the result in JSON format: {"question":" coresponding question to the sql provided"}

Figure 8: Prompt for question generation

### An example of test set for logical reasoning

**question**

Query the customer with the highest average overdue days under married status, whose latest loan was applied on a weekend, and provide the customer's gender and total income.

**Gold SQL**

SELECT app.NAME_FAMILY_STATUS, app.CODE_GENDER, app.AMT_INCOME_TOTAL, app.SK_ID_CURR, MAX(average_days_overdue) AS max_avg_overdue FROM (SELECT bureau.SK_ID_CURR, AVG(bureau.CREDIT_DAY_OVERDUE) AS average_days_overdue FROM bureau GROUP BY bureau.SK_ID_CURR) AS sub_bureau JOIN application_train AS app ON sub_bureau.SK_ID_CURR = app.SK_ID_CURR WHERE app.NAME_FAMILY_STATUS = 'Married' AND WEEKDAY_APPR_PROCESS_START IN ('Saturday', 'Sunday') GROUP BY app.SK_ID_CURR, app.NAME_FAMILY_STATUS, app.CODE_GENDER, app.AMT_INCOME_TOTAL ORDER BY max_avg_overdue DESC LIMIT 1;

**Logical Reasoning Steps**

**1. Calculate Average Overdue Days**
First, we need to calculate the average overdue days for each customer in the bureau table. This can be achieved with the following subquery:
SELECT bureau.SK_ID_CURR, AVG(bureau.CREDIT_DAY_OVERDUE) AS average_days_overdueFROM bureauGROUP BY bureau.SK_ID_CURR

**2. Filter and Join Tables**
We need to join the results of the above subquery with the application_train table and filter based on marital status and loan application date:
- Marital status is 'Married': app.NAME_FAMILY_STATUS = 'Married'
- Loan application date is on the weekend: WEEKDAY_APPR_PROCESS_START IN ('Saturday', 'Sunday')

**3. Group and Sort**
After joining and filtering, we need to group by customer ID, marital status, gender, and total income, and find the customer with the highest average overdue days.

Figure 9: An example illustration of test set for logical reasoning