# OpenReview forum: "Octopus: An Auto-Generated Multidimensional Fine-Grained Benchmark for Evaluating Text-to-SQL Systems"
_ICLR.cc/2026/Conference — ICLR 2026 Conference Withdrawn Submission_

### Official Review · Reviewer_t6kX · 2025-10-26

**Soundness:** 2
**Presentation:** 2
**Contribution:** 2
**Rating:** 2
**Confidence:** 4

**Summary:**

The paper introduced OCTOPUS, a multidimensional benchmark for evaluating text-to-SQL models, which is fully auto-generated and contains more complex queries and multi-turn Q&A. The author also proposed comprehensive evaluation metrics across four dimensions, accuracy robustness, interactivity, and generalization.

**Strengths:**

S1. Use the AST to abstract the SQL template, which is useful in extrapolating/generalizing the SQL clause.

S2. Including Multi-turn Q&A Generation is interesting and useful given the way the chatbot is being used.

S3. Clever design of the revert process, construct SQL and then generate natural language question.

S4. Relatively comprehensive metrics.

**Weaknesses:**

W1. Didn't have any sort of human measurement of the quality of the synthetic dataset, which makes me question the final quality and liability of the dataset.

W2. No measure of fidelity, how realistic the question compared to the real SQL question. The distribution of real data can be very different.

W3. Seems GPT4 has difficulty understanding the nested SQL structure, makes me wonder the choice of the model. The justification of the design choice is not convincing. What about reasoning models like o3, etc.?

W4. The paper discussed the datasets that are relative easy but didn't touch on hard ones like Spider2.0, and agentic baselines which tend to perform better on complex tasks.

**Questions:**

Q1. The real queries can be noisy, and don't use the exact value or column name in the database. How does the method account for that?

Q2. Any analysis around the SQL operation coverage? Proportion of each type of operations

Q3. Any difficulty level attached to each query? When reporting metrics, can be grouped by difficulty level.

Q4. Can you elaborate more on the challenges in generating multi-turn QA data? And how the proposed method is more advanced?

Q5. Any efficiency measurement?

---

### Official Review · Reviewer_aYLs · 2025-10-30

**Soundness:** 2
**Presentation:** 2
**Contribution:** 2
**Rating:** 2
**Confidence:** 4

**Summary:**

This paper proposes OCTOPUS, an automatically generated multidimensional benchmark for Text-to-SQL evaluation. It uses a GPT-4–based pipeline to produce question–SQL pairs across multiple databases and evaluation dimensions (accuracy, robustness, interactivity, generalization). The benchmark aims to replace human-labeled datasets with scalable automatic generation.

**Strengths:**

1. Ambitious attempt to fully automate Text-to-SQL benchmark generation with multi-dimensional evaluation.
2. Clear pipeline design and comprehensive metrics.
3. Large-scale dataset with low generation cost and reproducible workflow.

**Weaknesses:**

1. **Question-generation bias and lack of distribution validation:** The entire benchmark relies on GPT-4 to generate natural language questions from SQL templates without any comparison against real user queries or empirical text distributions. This introduces strong prior bias toward GPT-4’s linguistic and semantic style, which may not reflect actual user behavior. The paper does not provide any quantitative analysis, such as perplexity or embedding similarity between generated and real user questions, nor any human evaluation to verify representativeness. Consequently, the “diversity” and “authenticity” of questions remain unproven, threatening the benchmark’s validity.

2. Although the authors fine-tune CodeLlama, they omit more recent and stronger code-focused LLMs such as QwenCoder, which have demonstrated superior SQL understanding. This omission undermines the credibility of the experimental evaluation and prevents a fair assessment of current state-of-the-art systems. Using older models makes the reported results less meaningful for this year's performance comparison.

3. The paper argues that human-labeled datasets suffer from low SQL complexity and limited scalability, but this statement ignores the existence of advanced manually curated datasets such as Spider 2.0 that already include very complex multi-table and domain-specific queries. Without a systematic comparison to justify the claim, this argument appears overstated and weakens the motivation for a purely auto-generated approach.

4. While the proposed workflow is well-engineered, its workflow is similar to established pipelines from prior works such as Omni-SQL. The combination of existing techniques rather than fundamentally new algorithms limits the conceptual novelty.

**Questions:**

1. How did authors verify that GPT-4–generated questions reflect real user intent and linguistic diversity?
2. Did authors consider external human or model-based verification to cross-check GPT-4’s scoring?
3. How much overlap exists between the automatically generated SQLs and the templates or logic seen in prior datasets?
4. Authors still use EM as evaluation, which is the same as some previous works, such as Spider. I just wonder how authors evaluate the difference given the complexity authors claimed?

---

### Official Review · Reviewer_xEBw · 2025-10-31

**Soundness:** 2
**Presentation:** 2
**Contribution:** 2
**Rating:** 4
**Confidence:** 3

**Summary:**

The authors introduce OCTOPUS, a new benchmark for text-to-SQL systems, designed to address the limitations of existing datasets, such as high manual creation cost, low SQL complexity, and narrow evaluation scope. The primary contribution is a automated pipeline that generates complex question-SQL pairs. This pipeline uses SQL templates and a sampling algorithm to create SQL queries, then employs GPT-4 to translate these queries into natural language and subsequently filter for high-quality pairs. The resulting benchmark contains over 10,000 complex pairs and 10,000 multi-turn dialogues across 74 databases.

**Strengths:**

1. The paper addresses a clear and important problem: the need for more complex, scalable, and comprehensive benchmarks for text-to-SQL.
2. The automated generation pipeline is a valuable contribution.
3. The benchmark's design is comprehensive, incorporating multidimensional metrics that test aspects beyond simple accuracy.

**Weaknesses:**

1. The "Question-SQL Pair Selection" step relies heavily on GPT-4 as an automatic evaluator. This raises concerns about whether the benchmark evaluates general text-to-SQL capability or, to some extent, how well other models can align with GPT-4's internal biases.
2. There are various of typos. For example, at the end of the introduction section, the authors wrote that "... The main contributions of our paper are summarized as follows:", but there are no contribution listed behind. This is not an almost harmless issue. Besides, the header of table 3 appears to have layout issues.

**Questions:**

1. Given the small sample size of 100 for the evaluator validation, what steps could be taken to more robustly prove that the GPT-4 filter is a reliable proxy for human judgment across the entire dataset?
2. It would be helpful to provide an analysis of the discarded samples (those scoring < 80 ) to show what patterns GPT-4 is penalizing.
3. The selection of baseline models, including Qwen1.5, appears somewhat dated for a submission targeting 2026. Would the authors consider evaluating some newer models to provide a more current assessment of the benchmark's difficulty?

---

### Official Review · Reviewer_Vg3k · 2025-10-31

**Soundness:** 2
**Presentation:** 2
**Contribution:** 2
**Rating:** 2
**Confidence:** 4

**Summary:**

This paper introduces OCTOPUS, a comprehensive benchmark to address the limitations of existing human-annotated Text-to-SQL benchmarks, such as poor SQL complexity, one-query formats, and non-scalability. OCTOPUS is a fully auto-generated, multidimensional benchmark that evaluates Text-to-SQL systems on four important dimensions, including accuracy, robustness, interactivity, and generalization, on the basis of 9 first-level and 18 second-level metrics. The benchmark itself comprises 10,885 complicated question–SQL pairs and 10,874 multi-turn dialogue from 74 heterogeneous public databases from multiple domains (such as finance, sports, medicine). Among its advancements is its data creation pipeline that involves combining structured SQL template sampling, GPT-4-based SQL-to-natural-language translation, and LLM-quality scoring to ensure semantic correctness, fluency, and diversity. Experimental evaluations on cutting-edge LLMs confirm that the best models are not able to offer satisfactory performance across all evaluation metrics.

**Strengths:**

1. This paper introduces a fully automated data generation pipeline, combining SQL template sampling, GPT-4–based question generation, and LLM-driven quality filtering, which dramatically reduces human effort and ensures high scalability and reproducibility.

2. It provides a multidimensional and fine-grained evaluation framework, covering four critical dimensions, including accuracy, robustness, interactivity, and generalization, through 9 first-level and 18 second-level metrics, allowing for a comprehensive and systematic assessment of Text-to-SQL systems.

3. OCTOPUS achieves unprecedented dataset diversity and complexity, incorporating 10,885 question–SQL pairs and 10,874 multi-turn dialogues across 74 real-world databases, which better reflect practical application scenarios.

**Weaknesses:**

1. The motivation of this paper is not clear or convincing. A lot of benchmarks (BIRD, SPIDER-2, KaggleDBQA, BIRD-INTERACT) have been proposed to simulate more practical and challenging real-world text-to-SQL scenarios. However, this paper doesn't include them as competitors. What are the differences, and what is the advantage of proposing a new benchmark?

2. The construction of this dataset heavily relies on LLMs. How could you ensure that the generated questions and ground-truth SQLs are correct, reliable and comprehensive?

3. The writing of this paper is poor. This paper is more like a project report rather than a research paper since there are too many grammatical and formatting errors in this paper. For example, there is a clear format error in Table 3.

4. The efficiency is the key challenge in the text-to-SQL task; however, this benchmark overlooks this key aspect and focuses on some meaningless challenges that are impractical in real-world scenarios.

5. OCTOPUS exhibits notable weaknesses in metric analysis and interpretability. Although the benchmark introduces 9 first-level and 18 second-level metrics, the paper provides limited in-depth analysis of how these metrics interact, correlate, or individually affect model performance.

6. Most evaluations rely on aggregate execution accuracy, leaving the diagnostic value of finer-grained dimensions, such as logical reasoning complexity, robustness to perturbations, and SQL dialect generalization, underexplored. The study reports quantitative results but lacks qualitative or case-based analysis to explain failure patterns across different metrics, making it difficult to identify the root causes of poor model performance.

**Questions:**

1. What is the quality of the generated questions, SQLs and question-SQL pairs?

2. How effectively do the proposed first-level and second-level metrics capture distinct aspects of Text-to-SQL performance, and to what extent do they overlap or correlate in practice?

3. Why do current models fail on specific dimensions such as logical reasoning, robustness, or SQL dialect generalization, and what qualitative evidence can explain these weaknesses beyond aggregate accuracy scores?

---

### Note · Authors · 2025-11-14

I have read and agree with the venue's withdrawal policy on behalf of myself and my co-authors.